# Quality of Life and Post-Surgical Complications in Patients on Chronic Antiplatelet Therapy with Proximal Femur Fracture: 12-Month Follow-Up after Implementing a Strategy to Shorten the Time to Surgery

**DOI:** 10.3390/jcm12031130

**Published:** 2023-01-31

**Authors:** Angela Merchán-Galvis, Rafael Anaya, Mireia Rodriguez, Jordi Llorca, Mercé Castejón, José María Gil, Angélica Millan, Verónica Estepa, Elena Cardona, Yaiza Garcia-Sanchez, Ana Ruiz, Maria Jose Martinez-Zapata

**Affiliations:** 1Public Health and Clinical Epidemiology Service—Iberoamerican Cochrane Centre, IIB Sant Pau, 08025 Barcelona, Spain; 2Department of Social Medicine and Family Health, Universidad del Cauca, Popayan 190003, Colombia; 3Anesthesiology Service, Hospital de la Santa Creu i Sant Pau, 08025 Barcelona, Spain; 4Anesthesiology Service, Xarxa Assitencial Universitària de Manresa, 08243 Barcelona, Spain; 5Orthopedic and Traumatology Surgery Service, Hospital de la Santa Creu i Sant Pau, 08025 Barcelona, Spain; 6Anesthesiology Service, Hospital de la Vall d’Hebron, 08035 Barcelona, Spain; 7Orthopedic and Traumatology Surgery Service, Hospital Universitari Vall d’Hebron, 08035 Barcelona, Spain; 8Anesthesiology Service, Hospital Clinic de Barcelona, 08036 Barcelona, Spain; 9Centro de Investigación Biomédica en Red de Epidemiología y Salud Pública (CIBER of Epidemiology and Public Health), 28029 Madrid, Spain

**Keywords:** femur fracture, antiplatelet drugs, randomized clinical trial

## Abstract

Background: We evaluated a strategy to shorten the time from admission to surgery in patients with proximal femur fractures on chronic antiplatelet therapy. We reported a 12-month follow-up on complications and quality of life (QoL). Methods: Multicentre, open-label, randomized, parallel clinical trial. Patients were randomized to either early platelet function-guided surgery (experimental group) or delayed surgery (control group). Medical and surgical complications and QoL (EQ-5D-5L questionnaire) were assessed during the hospital stay, and after hospital discharge at 30 days, and 6 and 12 months. Results: From 156 randomized patients, 143 patients underwent surgery. The mean age was 85.5 (7.8) years and 68.0% were female. After hospital discharge, 5.7% of patients had surgical wound complications and 55.9% had medical complications, with 42.7% having serious adverse events. QoL improved significantly after surgery, with the best scores at the six-month follow-up. The overall mortality was 32.2%. There were no differences between early and delayed surgery groups in any assessed outcomes. Conclusion: It seems safe to reduce the time of surgery under neuraxial anaesthesia in patients with hip fractures on chronic antiplatelet therapy by platelet function testing. QoL in particular improves in the first six months after surgery.

## 1. Introduction

In developed countries, the incidence of proximal femur fractures in the elderly population has increased. This group of patients had chronic underlying pathologies, mainly of cardiovascular origin with chronic anti-platelet drug use. Associated comorbidities and chronic treatments increase perioperative complications in elderly patients, especially those undergoing proximal femur fracture surgery.

Early surgery (within 24 to 48 h) has been reported to have a positive effect on morbidity, hospital stay, perioperative complications, and mortality associated with a proximal femoral fracture [1,2,3,4,5,6,7,8,9]. This appears to apply to patients on chronic antiplatelet therapy [10,11,12].

Before performing hip fracture surgery, a complete assessment of the patient must be made. This includes regular medication revision, stabilization of the patient’s pathologies, and choosing an anaesthetic technique that will be most beneficial to the patient. At the moment there is little evidence on the best type of anaesthesia that would reduce perioperative and long-term complications in patients on chronic antiplatelet therapy [8,13,14,15,16]. Neuraxial anaesthesia has a potential risk for bleeding and causing epidural hematoma in these patients [17,18,19]; therefore, it is recommended to suspend antiplatelet therapy for three–seven days before surgery [19,20,21,22,23,24]. In contrast, patients receiving neuraxial anaesthesia are considered to have fewer thromboembolic events, acute kidney injury, pulmonary complications, and analgesic management is easier than general anaesthesia management [25,26,27,28,29]. However, a recent randomized clinical trial (RCT) developed in China showed no differences in delirium between spinal anaesthesia and general anaesthesia in the elderly with a hip fracture [30]. More recently, a RCT developed in the United States showed that spinal anaesthesia was associated with more pain in the first 24 h after surgery and more prescription analgesic use at 60 days compared with general anaesthesia [31]. However, patient satisfaction was similar in both groups [31].

As an alternative, we developed a clinical trial that evaluated an individualized strategy according to the platelet function test to shorten the time from admission to surgery without increasing perioperative complications in elderly patients with proximal femur fracture on chronic antiplatelet therapy [32,33]. This article reports the results on quality of life (QoL) and complications at 12 months after hospital discharge in these patients. 

## 2. Materials and Methods

We followed the recommendations of the Patient-Reported Outcomes (PRO) extension of the CONSORT guide for reporting this clinical trial [34].

This is a multicentre, open-label, randomized, parallel clinical trial, performed in four Spanish hospitals. The inclusion criteria were: Adult patients over 18 years of age with a proximal femur fracture and treatment with chronic antiplatelet agents such as cyclooxygenase inhibitors (acetylsalicylic acid (ASA) > 100 mg/day or triflusal > 300 mg/day) or P2Y12 receptor inhibitors (any dose of clopidogrel, prasugrel, ticagrelor or ticlopidine). Patients underwent orthopaedic surgery with neuraxial anaesthesia. All patients signed the informed consent before participation. Patients were recruited between 26 September 2017, to 5 December 2020. Patients with multiple or pathological fractures, treatment with vitamin K antagonists or new oral anticoagulants, and congenital or acquired coagulopathy were excluded. After hospital discharge, the information was collected through telephone calls to the patients or caregivers and a review of medical records. The data was collected on the electronic database platform Clinapsis^®^ (https://www.clinapsis.com/index.php/auth/login, accessed on 25 January 2023).

### 2.1. Interventions

On admission, antiplatelet medication was discontinued, and patients were randomized to early surgery (experimental group) or delayed surgery (control group), in both cases with neuraxial anaesthesia.

Experimental group: Platelet function was measured on admission to the emergency room to assess the anaesthetic safety of taking the patient to early surgery, having a minimum threshold of 80 × 10^9^/L of functional platelets, in which case, surgery was scheduled in the following 24 h. When the patient had <80 × 10^9^/L of functional platelets, platelet function was measured daily to verify the opportune moment to schedule surgery. If the minimum platelet threshold was not reached on the third day, surgery was scheduled following the margin of safety established for each antiplatelet drug as specified in the control group.

Control group: Surgery under neuraxial anaesthesia was performed following the safety margin of each antiplatelet agent, being 3 days for ASA > 100 mg/day and triflusal > 300 mg/day, 5 days for clopidogrel and ticagrelor, 7 days for prasugrel and 10 days for ticlopidine [21,22,23,24].

Orthopaedic surgeons acted in accordance with their usual clinical practice. Surgical technique, transfusion, and perioperative antibiotic and antithrombotic medication followed the protocol of each hospital. 

### 2.2. Outcomes

Medical and surgical complications and quality of life (QoL) assessments were collected during hospital stay and after hospital discharge, at 30 days, 6 and 12 months. 

Among the surgical complications, the presence of infection, dehiscence and/or hematoma of the surgical wound, dislocation of the prosthesis and/or surgical reintervention were evaluated. A medical complication was defined as any change in health status that required treatment or hospital admission, such as infections, changes in mental status, cardiovascular disease and/or death. Serious adverse events were defined as any event that caused death, was life-threatening, required hospitalisation, prolonged existing hospitalisation, or caused permanent or significant disability.

QoL was measured using the validated Spanish version of the generic questionnaire EQ-5D-5L (EuroQoL) [35], consisting of 2 pages: the EQ-5D descriptive system and the EQ visual analogue scale (EQ-VAS) [36]. The EQ-5D is a descriptive system with five domains (mobility, self-care, regular activities, pain/discomfort, and anxiety/depression) divided into five levels of severity (no problems, some problems, moderate problems, extreme problems, or unable) rated on an ordinal scale from 1 to 5 (where 1 indicates that there is no problem and 5 unable). We used the published index values of Spain’s population for calculating the EuroQol-5D-5L index values of the patients of the study [37]. The EQ-VAS performs a global assessment of the patient’s health through a vertical visual analogical scale that ranges from 0 to 100 (best score). The patient was asked about how his/her health had been on the day of the interview.

The instrument was applied at six different times after the fracture: before surgery, at 5 days, 1, 6- and 12-months post-surgery, considering the basal value of the QoL before the fracture obtained at the time of enrolment. Follow-up was done face-to face during the hospital stay and by telephone after discharge. The questionnaire was answered by the patient or a relative/caregiver.

### 2.3. Ethics

The study was approved by the ethics committee of each participating centres and registered in clinicaltrials.gov (NCT03231787). The protocol and perioperative results have already been published [32,33].

### 2.4. Statistical Analysis

Complications (except mortality) and QoL were analysed per protocol, considering all randomized patients who underwent surgery. Mortality was analysed by the intention-to-treat (ITT), including all randomized patients.

A descriptive and comparative analysis was made between the evaluated interventions of the medical and surgery-related events presented during the 12-month follow-up.

The qualitative variables are presented as frequencies and percentages, and for the quantitative variables the mean with the standard deviation (SD) or the median and the IQR according to normality were calculated. To compare the results between groups, the Chi square test was applied.

Mortality was reported as a cumulative incidence proportion. Survival was calculated from the time of hospital admission to 12 months after surgery by group. The survival differences were calculated by Kaplan-Meier method using the Log rank test.

The analysis of QoL included all patients who underwent surgery and had preoperative QoL and at least one postoperative measurement. We imputed missing data with the following criteria:When there was no data prior to the fracture or surgery, the average of all the patients of the corresponding evaluation was imputed.If a measurement was missing between two points of the same patient, the measurements before and after the missing data were averaged.If tracking was lost, the last recorded value was forward dragged.When the patient died at the point of measurement, 0 was imputed and no further follow-ups were recorded.

We evaluated QoL throughout the study, from baseline to the end of follow-up. For changes in the VAS and EQ-5D-5L scores, a two-way ANOVA was performed. The factors were group (intervention and control), time (baseline, at day 5, and at 1, 6, and 12 months after surgery), and the interaction between them. The ANOVA was performed using a general linear model (GLM) procedure.

We used a significance level of 0.05. All statistical analyses were performed with IBM Corp. Released 2017. IBM SPSS Statistics for Windows, Version 25.0.; IBM Corp., Armonk, NY, USA.

## 3. Results

This study included 156 patients who were randomized to an intervention (*n* = 78) and control group (*n* = 78). Before surgery, one patient in the intervention group and two in the control group removed their consent; three patients in the control group were withdrawn for not meeting some inclusion criteria. This report corresponds to the remaining 150 patients; 77 participated in the experimental group and 73 in the control group (Figure 1). Six randomized patients died before surgery (two in the experimental group and four in the control group), and in one patient the surgery was suspended in the control group. Therefore, 143 patients underwent surgery. The mean age was 85.5 years (7.8) and 68.0% were female. The mean time from admission to surgery was 2.8 days in the intervention group and 5.3 days in the control group (Table 1).

Among the 143 patients that underwent surgery, four patients in each group died in-hospital. Therefore, 135 patients were discharged alive. After hospital discharge and during the 12-months of follow-up, patients presented a median of two complications per patient (from 1 to 6) and a median of appearance of 110 days (from 1 to 350). In total, eight (5.7%) patients presented surgical wound complications, 80 (55.9%) presented medical complications, and 61 (42.7%) presented serious adverse events without differences between groups. The most common complications after discharge were death (34 patients), urinary tract infection (24 patients) and cardiovascular diseases (13 patients) (Table 2; Figure 2).

In an ITT analysis, the overall mortality of the study was 32.2% (*n* = 48). The mean survival time was 275.9 days (11.6) with no statistical difference between groups (*p* = 0.561) (Figure 3). Post-discharge, the mortality rate was 23.8% (*n* = 34) with no difference between groups (*p* = 0.521).

In the QoL analysis, 128 (89.5%) of the 143 patients that underwent surgery were included. The global assessment of quality of life measured with the VAS prior to the fracture was 63.4 and 39.0 after the fracture at hospital admission. In successive follow-ups after surgery, VAS improved significantly (*p* < 0.0001) but without reaching baseline values and without differences between groups (Appendix A).

The QoL index evaluated with the EQ-5D-5L instrument had an average of 0.555 before the fracture and decreased to −0.349 before surgery. After surgery, patients had a progressive and significant increase in QoL index scores, reaching up to 0.340 at 12-month follow-up (*p* < 0.0001), with no difference between groups (Appendix A).

Figure 4 shows the evolution in the mean index of EQ-5D-5L (Figure 4A) and EQ-5D-VAS (Figure 4B) during the 12-month follow-up.

## 4. Discussion

Accelerating the time to surgery with neuraxial anesthesia in patients with a proximal femur fracture chronically medicated with antiplatelet drugs did not result in significant differences in quality of life nor in the incidence of complications when compared with those receiving delayed surgery at 12-month follow-up.

The chronic use of antiplatelet agents is an important risk factor for complications such as bleeding and mortality in patients undergoing surgery. Therefore, it is recommended to suspend the medication prior to the surgical procedure, especially if neuraxial anesthesia is used due to its high risk of bleeding [19,23,38]. However, in recent years, various studies have found that stopping platelet antiaggregant for prolonged periods (1 week) can lead to thromboembolic complications, infections, and longer hospital stays. As a result, different systematic reviews have evaluated the use of antiplatelet drugs in hip fracture surgery, finding that early surgery (<3 days) regardless of the anesthetic technique, reduces hospital stay and the risk of mortality [10,12,39,40]. In a previous publication, our study showed that it is possible to safely accelerate surgery with neuraxial anaesthesia by implementing a platelet function test, shortening hospital stay without increasing complications [31]. The 12-month follow-up shows that these results are maintained over time.

The average age of around 80 years is consistent with global reports on the incidence of proximal femur fracture in patients with cardiovascular comorbidity and a higher frequency in women [41,42,43,44]. Most patients in this study had cardiovascular comorbidity, which corresponds to ASA 3 functional status in almost 90% of the patients and could explain the relatively higher mortality compared to other studies [39,41,44,45,46,47,48,49].

Most complications in our study occurred during the first 4 months after surgery, with no differences between compared groups. The most frequent postoperative medical complications were urinary tract infections and cardiovascular events, which occurred after 60 and 100 days from randomization, respectively. These complications are commonly reported in the literature [40,43,46,50] and generally related to the patient’s previous health status with no relationship between the type of anaesthesia or surgery used, but with the time from admission to surgery [39,46,48]. The strength of recommendation is moderate for implementing the surgery in the first 24–48 h after admission to the hospital [9]. However, there is variability regarding the postoperative results related to the type of anaesthesia applied to the patients. In this sense, various systematic reviews have attempted to establish the benefits of one anaesthetic technique over the other without reaching conclusive recommendations due to the weak evidence based mainly from uncontrolled or retrospective studies, which suggest that regional anaesthesia could present fewer complications and perioperative mortality [27,29]. However, when systematic reviews only included randomized controlled trials, no differences were observed between both types of anaesthesia [28].

Quality of life has been under investigated in patients with proximal femur fracture and chronic antiplatelet therapy. This study found that, after surgery, the general assessment of health status and the patient’s index of quality of life improved significantly with respect to the moment of the fracture, especially in the first six months after the intervention; however, they did not reach the pre-trauma baseline values. This is consistent with various studies on quality of life in proximal femur fractures [51]. The suboptimal improvement could be explained by advanced age, previous patient comorbidities and the lack of long-term physiotherapy measures [52,53].

This study stands out for its design, as it was a multicentre randomized clinical trial with prolonged patient follow-up. It contrasts with most published primary studies on complications and quality of life that are observational and retrospective. However, the relatively small sample size has been a limitation. Studies with a larger number of patients would be required to corroborate the findings of this study.

## 5. Conclusions

This study suggests that it is possible to reduce the time to surgery under neuraxial anaesthesia in patients with hip fracture receiving chronic antiplatelet therapy using a platelet function test, without increasing post-surgical complications, serious adverse events, or mortality, and not worsening their quality of life up to 12 months after surgery.

## Figures and Tables

**Figure 1 jcm-12-01130-f001:**
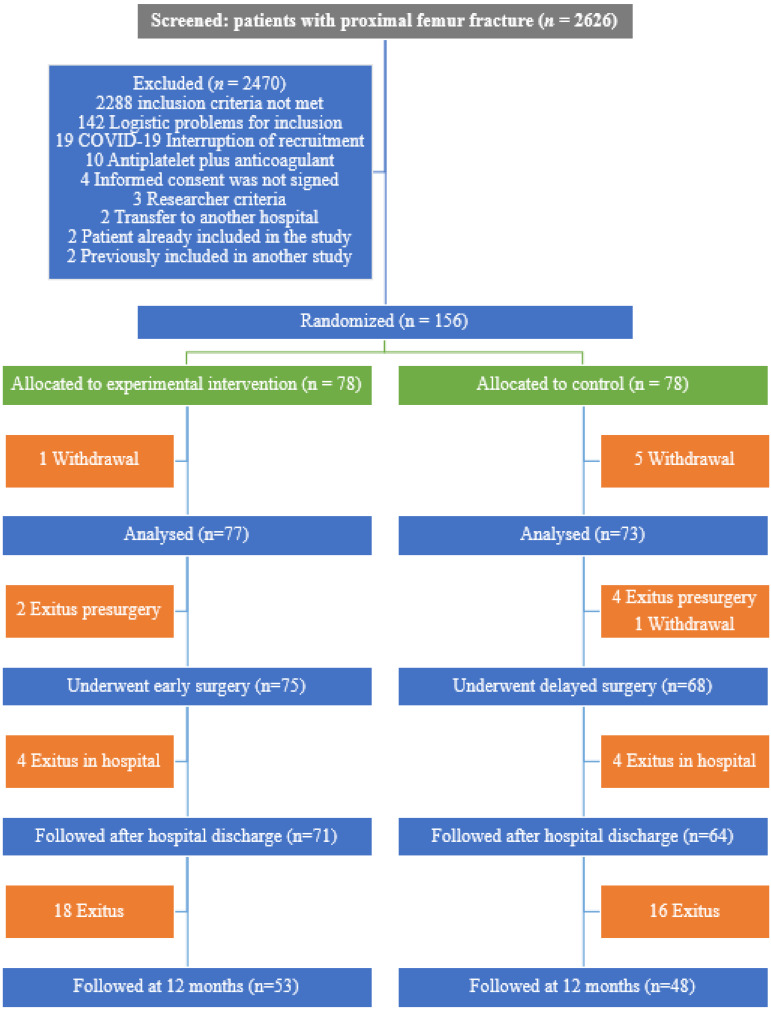
Selection of patients included in the study.

**Figure 2 jcm-12-01130-f002:**
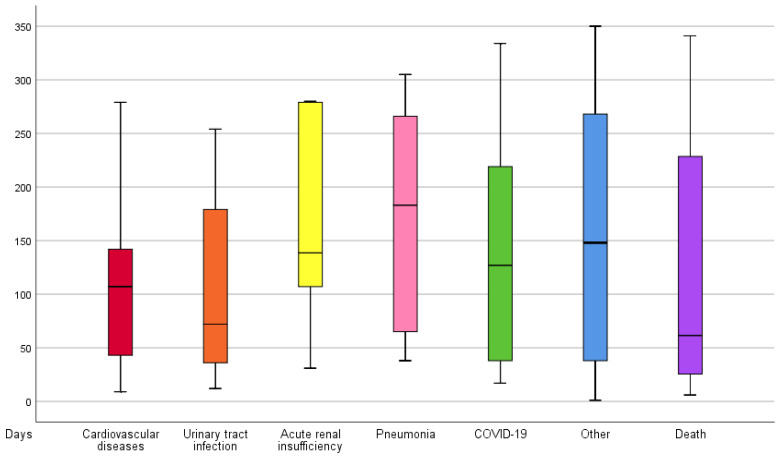
Time to complications after hospital discharge in days.

**Figure 3 jcm-12-01130-f003:**
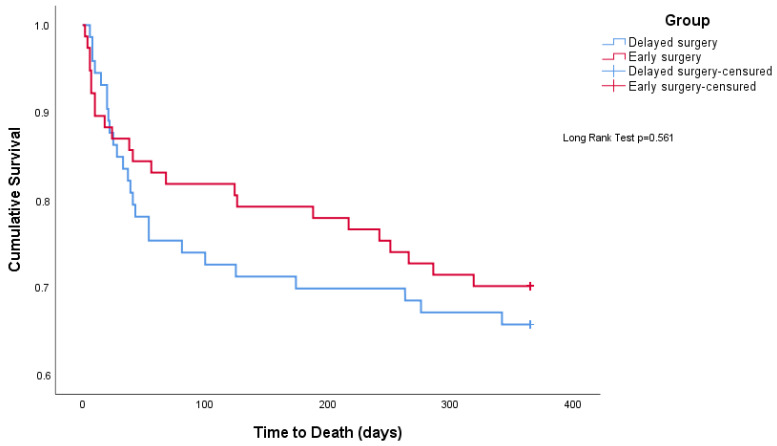
Patients’ survival by patients by group.

**Figure 4 jcm-12-01130-f004:**
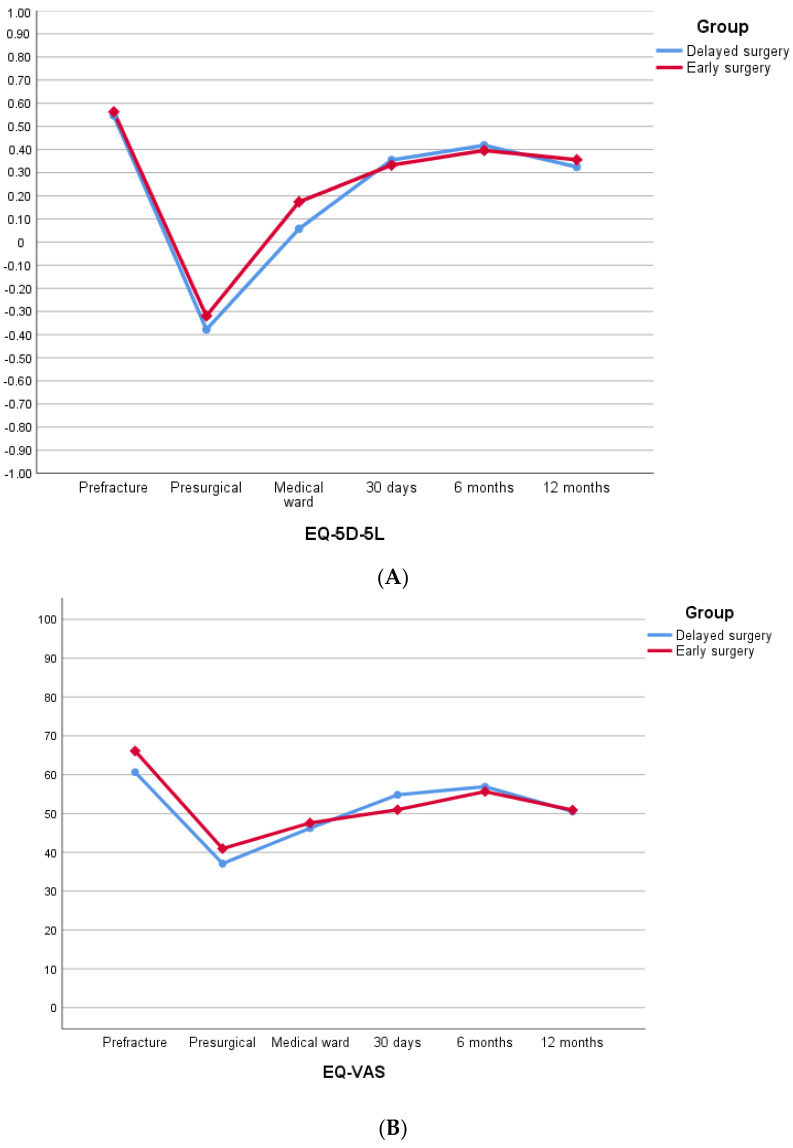
(**A**) EQ-5D-5L index assessment at different follow-up times. (**B**) EQ-5D VAS assessment at different follow-up times.

**Table 1 jcm-12-01130-t001:** Characteristics of the study population randomized and did not withdraw their consent before surgery.

		Early Surgery *n* = 77		Delayed Surgery *n* = 73		Total *n* = 150		*p*
		*n*	(%)	*n*	(%)	*n*	(%)	
Centre	Hospital 1	39	(50.6)	37	(50.7)	76	(50.7)	0.980
	Hospital 2	17	(22.1)	15	(20.5)	32	(21.3)	
	Hospital 3	15	(19.5)	14	(19.2)	29	(19.3)	
	Hospital 4	6	(7.8)	7	(9.6)	13	(8.7)	
Gender	Male	21	(27.3)	27	(37.0)	48	(32.0)	0.136
	Female	56	(72.7)	46	(63.0)	102	(68.0)	
Age	Mean (SD)	85.0	(8.7)	86.1	(6.8)	85.5	(7.8)	0.358
Surgery	No	2	(2.6)	5	(6.8)	7	(4.7)	0.200
	Yes	75	(97.4)	68	(93.2)	143	(95.3)	
Type of femur fracture	Intracapsular	34	(45.3)	37	(54.4)	71	(49.7)	0.180
	Extracapsular	41	(54.7)	31	(44.6)	72	(50.3)	
Time to surgery	Mean (SD)	2.8	(1.7)	5.3	(2.1)	3.9	(2.3)	0.000
American Society of Anaesthesiologists’ score	1	1	(1.3)	0	(0.0)	1	(0.7)	0.473
	2	3	(4.0)	6	(8.7)	9	(6.3)	
	3	67	(89.3)	58	(84.1)	124	(86.7)	
	4	4	(5.4)	5	(7.2)	9	(6.3)	
Type of surgery	Osteosynthesis	46	(61.3)	36	(52.9)	82	(57.3)	0.199
	Arthroplasty	29	(38.7)	32	(47.1)	61	(42.7)	

**Table 2 jcm-12-01130-t002:** Medical and surgical complications after hospital discharge per patient.

	Early Surgery *n* = 75		Delayed Surgery *n* = 68		Total *n* = 143		*p*
	*n*	(%)	*n*	(%)	*n*	(%)	
Wound complications	4	(5.4)	4	(6.0)	8	(5.7)	0.584
Patients with medical complications	42	(56.0)	38	(55.9)	80	(55.9)	0.510
Death	18	(24.0)	16	(23.5)	34	(23.8)	0.521
Urinary tract infection	14	(18.7)	10	(14.7)	24	(16.8)	0.304
COVID-19	6	(8.0)	3	(4.4)	9	(6.3)	0.280
Pneumonia	5	(6.7)	5	(7.4)	10	(7.0)	0.582
Cardiovascular diseases	5	(6.7)	8	(11.8)	13	(9.1)	0.226
Acute renal insufficiency	1	(1.3)	5	(7.4)	6	(4.2)	0.085
Other *	17	(22.7)	17	(25.0)	34	(23.8)	0.477
Serious adverse events per patient	31	(41.3)	30	(44.1)	61	(42.7)	0.330

* Events with a frequency of less than 5 were grouped in other.

## Data Availability

The data presented in this study are available on request from the corresponding author. The data are not publicly available due to this possibility and were not included in the informed consent of the patient.

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
