# Peer review of "Quality of Life and Post-Surgical Complications in Patients on Chronic Antiplatelet Therapy with Proximal Femur Fracture: 12-Month Follow-Up after Implementing a Strategy to Shorten the Time to Surgery"

_jcm, 2023, doi:10.3390/jcm12031130_

Round 1

Reviewer 1 Report

No significant differences in clinical outcomes were found before early and delayed hip-fracture surgery in patients receiving antiplatelet therapy. Knowing the fact that the population is getting older, and that proportion of these patients will increase with time, it is encouraging to know that these patients can be operated early, without significant impact on postoperative mortality and QoL.

This way, we can expect shorter duration of hospital stay, earlier mobilization and I sincerely hope that some larger multi-national, multi-centric studies will show a reduced incidence rate of infectious complications.

Also, I would like to know if earlier surgery reduced the number of transfused RBC units (i expect yes, since many of these patients need to receive RBCs while waiting for surgery due to blood loss from fractured bone).

If the authors have the data, it would be nice if it could be added.. If not, never mind.

Author Response

Answer: Thanks very much for the comments.

The English language has been reviewed by a native.

We have a previous publication (Anaya 2021) with the number of transfusions; 70.6% of patients received at least one unit of red blood transfusion during the perioperative period, and there were no differences between groups. In the pre-operative period, there were no differences because of the low number of patients transfused. The hemoglobin pre-surgery was 10.9 in both groups (Table 4 of Anaya 2021).

(1) Anaya et al. Early Surgery with Neuraxial Anaesthesia in Patients on Chronic Antiplatelet therapy with a Proximal Femur Fracture: Multicentric Randomised Clinical Trial. J. Clin. Med. 2021, 10, 5371.

Reviewer 2 Report

The study is well designed and the paper is written to a high scientific standard. My only point I want to see improved is for the discussion and background. For example in germany there is now a law that ban hip fracture treatment after 24h. There is like the authors cited literature showing a disadvantage for treatment after 24 and 48 hours. So your time to surgery times are hard to understand in this context. Still there is evidence that regional anaesthesia might be benefical for geriatric patients. So both introduction and discussion must better describe this connection. Because there is already evidence from the surgical side that treatment within 24 hours despite anti-platelet therapy shows no disadvantages. So the benefit of regional anaesthesia has to balance out the disadvantages of the delay. 

Author Response

Answer: Thanks very much for the comments. We have included in the introduction and in the discussion the recent recommendations of the AAOS about performing surgery in the first 24-48h after hospital admission. We have also included information based on new recent publications about the controversy between neuraxial and general analgesia.